# Long-Time Evaluation of Solid-State Composite Reference Electrodes

**DOI:** 10.3390/membranes12060569

**Published:** 2022-05-30

**Authors:** Slim Blidi, Kim Granholm, Tomasz Sokalski, Zekra Mousavi, Andrzej Lewenstam, Ivo Leito, Johan Bobacka

**Affiliations:** 1Johan Gadolin Process Chemistry Centre, Laboratory of Molecular Science and Engineering, Åbo Akademi University, Henriksgatan 2, FI-20500 Turku-Åbo, Finland; s.blidi1700@abertay.ac.uk (S.B.); kgranholm@abo.fi (K.G.); zmousavi@abo.fi (Z.M.); alewenst@abo.fi (A.L.); jbobacka@abo.fi (J.B.); 2Faculty of Materials Science and Ceramics, AGH University of Science and Technology, Mickiewicza 30, 30-059 Krakow, Poland; 3Institute of Chemistry, University of Tartu, Ravila 14a, 50411 Tartu, Estonia; ivo.leito@ut.ee

**Keywords:** reference electrode, solid-state, long-time evaluation

## Abstract

In this study, the performance and long-time evaluation of solid-state composite (SSC) reference electrodes were investigated. The stability of all the SSC reference electrodes was continuously monitored by using potentiometry and electrochemical impedance spectroscopy methods over a period of several months. A multi-solution protocol was used to study the influence of the ionic strength of the sample solution, ion charge, and mobility, and the sample pH values on the performance of the reference electrodes. The SSC reference electrodes were used in the calibration of commercial indicator electrodes for different ions at different temperatures. The concentrations of K^+^, Na^+^, Ca^2+^, and Cl^-^ ions and pH values were measured in river water samples at different temperatures using the SSC reference electrodes. The obtained results for the same samples were compared with the results given by an independent laboratory specialized in routine water analyses. The agreement between the results was very good and even better than the case where commercial reference electrodes were used. Our study showed that the SSC reference electrodes exhibit good long-term stability and excellent performance, both in the calibrations and analyses of environmental samples.

## 1. Introduction

The reference electrode in an electrochemical measurement is indispensable in order to ensure an accurate, stable, and reproducible potential that is independent of the composition of the sample solution [1]. Nowadays, the most commonly used reference electrodes are silver/silver chloride (Ag/AgCl), and calomel (Hg/Hg_2_Cl_2_) conventional reference electrodes. An inner filling solution is the basic component in the construction of conventional reference electrodes, which may cause several problems and inconveniences when used. Such a construction is complicated and expensive to manufacture. It is also maintenance-intensive due to the necessity of refilling the inner solution and keeping the liquid junction clog-free. Additionally, it is difficult to miniaturize the electrode because the volume of the internal solution cannot be very small. Another drawback is that the electrode works only in upright position to ensure the contact between the sample solution and the inner reference part. A very important issue is the leakage of the inorganic salt from the internal solution into the sample, which may change the sample composition and lead to errors in the results. Furthermore, the electrode frit can be clogged due to fouling, and this makes the response of the electrode unreliable, especially in environmental and industrial applications. Several attempts have been made to alleviate the above-mentioned drawbacks and disadvantages. For instance, gel-like electrolyte [2,3,4], as well as a melted salt of potassium chloride [5], have substituted the inner filling solution. To minimize problems associated with the junction potential, junction-less reference electrodes were developed [6,7,8] or a low-resistance material was used as the electrode junction [9]. Lipophilic salts [10,11], conducting polymers [12], conducting polymers doped with pH buffering ligands [7], carbon nanotubes [13,14], and ionic liquids [15] were also used for the preparation of solid-state reference electrodes.

The comparative study of the reference electrodes with polymer-based membranes was published recently [16]. There are also studies of using solid-state reference electrodes in extreme conditions (high temperature, high salinity, and high pressure) [17,18].

In our previous works, we developed and studied analytical-quality solid-state reference electrodes based on a polymer/inorganic salt composite. The electrodes were prepared using chemical polymerization [19,20] or injection molding [21] methods. In this construction, a silver/silver chloride (Ag/AgCl) wire, embedded in the polymeric matrix, acts as a reference element. The composite acts both as an inorganic salt storage material and liquid junction contact with the sample solution at the same time. The electrodes exhibited as good as, and in many aspects, even better properties than high quality commercially available reference electrodes in terms of reduced leakage of inorganic salt, insensitivity to the matrix effects, and stability of the potential.

In this study, long-time evaluation of the solid-state composite reference electrodes is presented. The performance of the reference electrodes was tested in different synthetic solutions to assess the possible influence of pH, solution composition, as well as the concentration and mobility of ions. Stability tests were also performed to evaluate the suitability of the studied SSC reference electrodes for continuous, prolonged, and intensive usage. In addition, the electrodes were tested in real environmental samples with complex matrices.

## 2. Experimental

### 2.1. Reagents

Calcium chloride dihydrate (CaCl₂·2H₂O), Baker Analyzed^®^, p.a. of ≥99.0% and sodium bromide (NaBr), Baker Analyzed^®^, p.a. of ≥99.0% were purchased from J.T Baker, Deventer, Holland. Hydrochloric acid 30% (HCl), Suprapur^®^, potassium hydroxide (KOH) Pellets G.R. for analysis, p.a. of ≥85.0%, and sodium hydroxide (NaOH) Pellets G.R. for analysis EMSURE^®^ ISO, p.a. of 99.0% were purchased from Merck, Darmstadt, Germany, and pH buffers 4.01, 7.00 and 10.01 from Thermo Scientific, Waltham, MA, USA. Potassium chloride (KCl), p.a. of ≥99.0% from Fluka, Switzerland. Sodium bicarbonate (NaHCO_3_), p.a. of ≥99.5% was from Sigma, USA. Sodium chloride (NaCl), p.a. of 99.73% was from Fisher Scientific, United Kingdom. Vinyl acetate monomer (VAc, ≥99%), photo-initiator 2,2-dimethoxy-2-phenylacetophenone (DMPP, 99%), and lithium acetate (LiOAc, 98%) were purchased from Aldrich, St. Louis, Missouri, USA. Polyvinyl acetate (PVAc) powder was obtained from Wacker (Vinnapas B60 Finely ground). Polypropylene was obtained from Ultrapolymers Group, Lommel, Belgium. Pellets of (polypropylene + KCl) composite were prepared by Premix Oy. Distilled and deionized water (DW) (ELGA Purelab Ultra; resistivity 18.2 MΩcm) was used to prepare all solutions.

### 2.2. Preparation of the SSC Reference Electrodes

#### 2.2.1. SSC Reference Electrodes Prepared by Chemical Polymerization Method

The protocol for manufacturing SSC reference electrodes by chemical polymerization method (denoted as CP SSC) was described earlier [19,22]. Briefly, a mixture of polyvinyl acetate, potassium chloride, vinyl acetate monomer, and the photo initiator (DMPP) was poured into a plastic vial and an Ag/AgCl reference element was inserted in the centre of the vial keeping a suitable distance from the bottom of the vial. Afterwards, the polymerization was done using a UV lamp. After cutting the bottom of the vial, the resulted cylindrical electrode of the size 8 × 40 mm was conditioned in 0.1 M KCl solution. Three SSC electrodes (CP1–CP3) were prepared and used in the study.

#### 2.2.2. SSC Reference Electrodes Prepared Injection Molding Method

The protocol for manufacturing SSC reference electrodes by injection molding (denoted as IM SSC) was adapted from our previous work [21,22] with some minor changes. Prior to the injection process, an Ag/AgCl wire was placed in the injection molding form. Then, the polymer/salt composite was injected at 170 °C using the injection molder (JKV 105-2/100, Helsinki, Finland). The prepared cylindrical electrodes of the size 9 × 40 mm were then conditioned in 0.1 M KCl solution. Three SSC electrodes (IM1–IM3) were prepared and studied in this work.

### 2.3. Characterization Methods

#### 2.3.1. The Stability of SSC Reference Electrodes

The stability of the prepared electrodes was studied over a period of 25 weeks and 12 weeks for IM SSC and CP SSC reference electrodes, respectively. The IM SSC electrodes were monitored for a longer time than the CP SSC electrodes because the former ones take a longer time to be fully conditioned. Three identical electrodes of each type were immersed in an unstirred 10^−1^ M KCl solution, and the potential was recorded for five minutes against Ag/AgCl/3 M KCl/1 M KCl double junction reference electrode (RL 100 D/J Elmetron, Zabrze, Poland). The potential was recorded using the 16-channel milli-voltmeter (Lawson Labs. Inc., Malvern, PA, USA).

#### 2.3.2. Electrochemical Impedance Spectroscopy (EIS)

The electrochemical impedance spectroscopy measurements were performed in a conventional one-compartment three-electrode electrochemical cell using an Autolab General Purpose Electrochemical System (AUT20.FRA2-Autolab, Eco Chemie, B.V., Utrecht, The Netherlands) and the Autolab Frequency Response Analyzer (FRA) software. The impedance spectra of the electrodes were recorded in unpurged and unstirred 10^−1^ M KCl solution during 25 weeks and 12 weeks for IM SSC and CP SSC reference electrodes, respectively. A glassy carbon electrode and Ag/AgCl/3 M KCl reference electrode (6.0733.100 Metrohm, Herisau, Switzerland) were used as counter and reference electrodes, respectively. The frequency was scanned from 500 kHz to 0.1 Hz by recording 55 measuring points. The used excitation potential amplitude was *E*_ac_ = 10 mV and *E*_dc_ = 0 V.

#### 2.3.3. Multi-Solution Protocol

The multi-solution protocol (MSP) was carried out to investigate the effect of the nature and concentration of the electrolyte solution on the response of the selected commercial and SSC reference electrodes [19,21]. The response of SSC reference electrodes was monitored in solutions containing different cations (K^+^, Na^+^, Ca^2+^) and anions (Cl^−^, Br^−^, HCO_3_^−^). In addition, the effect of the solutions’ ionic strength on the electrode response was investigated by measuring the potential in equitransferent (KCl) and non-equitransferent (CaCl_2_) solutions with different concentrations. Furthermore, the effect of solution pH was investigated by performing the measurement in acidic (HCl) and basic (NaOH) solutions. The potential was recorded in each solution for five minutes against a double-junction reference electrode (800500U Orion Ross Ultra D/J, Thermo Scientific, Waltham, MA, USA). The SSC reference electrodes were rinsed with deionized water between different solutions. The obtained potential values were normalized taking the first recorded potential reading as a reference value. Two sequences (MSP1 and MSP2) were used during the multi-solution protocol. In the MSP1, the sequence of used solutions was as previously described [19,21]: 3 M KCl, deionized water, 10^−2^ M NaCl, 10^−2^ M KCl, 10^−1^ M KCl, 10^−2^ M HCl, deionized water, 3 M KCl, 10^−1^ M NaCl, 10^−1^ M KCl, 10^−1^ M NaBr, 10^−1^ M NaHCO_3_, 10^−3^ M KOH, 10^−2^ M HCl, and 3 M KCl. In MSP2, the solutions were used in the following order: 10^−1^ M KCl, 10^−1^ M CaCl_2_, 10^−4^ M CaCl_2,_ deionized water, 10^−1^ M HCl, 10^−4^ M HCl, deionized water, 10^−1^ M NaOH, and 10^−4^ M NaOH. The original MSP1 protocol was proposed for investigating reference electrodes stability in clinical applications. The MSP2 protocol was designed to study the influence of higher concentration of acids and bases and doubly charged cations on the potential stability of the SSC reference electrodes.

#### 2.3.4. Calibration Measurements

The prepared SSC reference electrodes were used in the calibration of different commercial ion-selective electrodes (ISEs). The used ISEs were Ca-ISE (6.0508.100), Na-ISE (6.0508.100) from Metrohm, Herisau, Switzerland), and Cl-ISE (Orion 9417BN), K-ISE (Orion 9300BN), and pH electrode (Orion 8101BNWP) from Thermo Scientific, USA. For comparison purpose, an Orion double-junction reference electrode was also used in this study. The calibrations were performed at 5 °C, 14 °C, and 18 °C using a thermostat (Lauda E100 Ecoline 20 Liter, Lauda-Brinkmann, Delran, NJ, USA), and at room temperature (22 ± 1 °C). All calibrations were done by automatic dilution of a stock solution using Metrohm Dosino 700 instruments equipped with burettes of 50 mL capacity (Herisau, Switzerland). For pH measurements, 4.01, 7.00, and 10.01 buffers were used.

As a preliminary step, the linear range was estimated visually from the electrodes’ calibration curves. Next, the linear regression was carried out in this range using the commercial software (Excel or Origin). In this way, the slope, intercept, standard errors, and the goodness of fit represented by coefficient of determination (R^2^) were obtained.

#### 2.3.5. Sample Measurements

Concentrations of K^+^, Na^+^, Cl^−^, Ca^2+^ ions, and pH were measured in river water samples collected from two different locations near Turku, Finland. The first sample (RW1) was taken from Aura River close to the Halistenkoski region. The second sample (RW2) was taken from the Savijoki stream. The measurements were carried out at different temperatures: 5 °C, 14 °C, 18 °C, and at room temperature (22 ± 1 °C) using the SSC and the commercial Orion Ross Ultra double-junction as reference electrodes. The water samples were stored in the fridge at 4 °C before measurements. The measurements were performed within four days after the sampling.

## 3. Results and Discussions

### 3.1. Potential Stability of the Electrodes

The long-time potential stability of IM SSC and CP SSC electrodes measured versus a commercial reference electrode is shown in Figure 1. The measurements were done daily for 5 min and the electrodes were kept in the conditioning solution between the measurements. As can be seen in Figure 1a, from the beginning until week 4, the potential value for the CP SSC electrode was ca. −0.3 mV, and after 4 weeks it was ca. −3 mV. The few millivolts potential difference is due to the fact that there is 3 M KCl in the commercial reference electrode, while the concentration of KCl in CP SSC reference electrode is slightly higher.

On the other hand, the results in Figure 1b show that, compared to the CP SSC, it took a longer time for the IM SSC electrode to be fully conditioned and for the potential to stabilize. This can be due to the difference in the polymer used, and the porosity and hardness of the polymer/KCl composite in these two cases. After about 15 weeks, the potential value was ca. −3 mV, which is similar to the value obtained for CP SSC electrode. Generally, the CP SSC and IM SSC reference electrodes showed an excellent long-term stability even after 12 and 25 weeks, respectively. The reproducibility of three identical CP SSC and IM SSC electrodes are illustrated in Figure 2 by measuring their potential for 5 min in 10^−1^ M KCl solution. For both types of electrodes, the reproducibility is within 1 mV, which is very good.

### 3.2. Electrochemical Impedance Spectroscopy (EIS)

The EIS measurements were performed in order to obtain the resistances of the SSC reference electrodes during the conditioning process. Typical impedance spectra of CP SSC and IM SSC electrodes recorded in 0.1 M KCl are shown in Figure 3. The resistance values of SSC reference electrodes were obtained from the real value of the impedance (Z’) at the lowest frequency (0.1 Hz).

Before conditioning, the resistance for both CP SSC and IM SSC electrodes was in the range of GΩ. During the conditioning, the resistance values were decreasing until a reasonably stable value of ca. 7 kΩ was obtained for CP SSC after 5 weeks (Figure 4a). In the case of IM SSC, a relatively stable resistance value of ca. 6 kΩ was obtained after 10 weeks of conditioning (Figure 4b). These results are in good agreement with the results from potential stability measurements shown in Figure 1.

### 3.3. Multi-Solution Protocol

*CP SSC electrodes*: In the MSP1, all CP SSC reference electrodes demonstrated excellent stability (± 1 mV) with all the tested solutions, as shown in Figure 5a. In the MSP2 shown in Figure 5b, the drastic variation in pH of the solution (i.e., hydrochloric acid as a strong acid and sodium hydroxide as a strong base) had a bigger influence on the response of the electrodes (± 2 mV).

*IM SSC electrodes*: In the MSP1 (Figure 6a), the potential stability was within ±1 mV except for the case of 10^−2^ M HCl and 10^−3^ M KOH solutions where it was between +2 and −6 mV. In the case of MSP2 (Figure 6b), the potential stability was worse than that of CP SSC electrodes within 5 and −6 mV. Therefore, it is clear that the IM SSC reference electrodes are more sensitive to harsh pH changes than CP SSC.

### 3.4. Calibration Measurements

During the potential stability test, the multi solution protocol, and the impedance measurements, the SSC reference electrodes (both CP and IM) were tested as indicator electrodes versus commercial reference electrodes. The ultimate test for the suitability of SSC electrodes as reference electrode was done by using them as reference electrodes in the calibration of commercial indicator electrodes, as shown below. The calibrations were done using CP SSC and IM SSC after 12 and 25 weeks of conditioning, respectively.

The K^+^, Na^+^, Ca^2+,^ Cl^−^, and pH commercial ISEs were used as indicator electrodes. The calibrations of ISEs were performed using three identical CP SSC or IM SSC reference electrodes at room temperature (22 ± 1 °C). The slope (*S*), standard potential (*E*^0^), and coefficient of determination obtained from the calibration curves using SSC electrodes are presented in Table 1 below.

As it can be seen from the results in Table 1, the calibration curves obtained with both CP SSC and IM SSC as reference electrodes show near-Nernstian slopes and stable standard potentials (The Nernstian slope is given by the product ln(10)·(RT/zF) and equals 59.16 mV at T = 298,16 K for z = 1). It can be concluded that the SSC electrodes functioned very well as reference electrodes.

### 3.5. Effect of the Temperature on the Nernstian Slope

The calibration curves presented earlier were obtained at room temperature (22 ± 1 °C). However, in environmental measurements of natural waters, the temperature is usually lower than room temperature. The temperature during the measurements affects the slope of an ion-selective electrode due to the (RT/zF) term in the Nernst equation. As the temperature decreases, the Nernstian slope will decrease. The theoretical ratio of Nernstian slopes at two different temperatures should be equal to the temperature ratio in kelvins (Equation (1)).
(1)$$\mathrm{Theoretical}\;\mathrm{ratio}\;\mathrm{of}\;\mathrm{Nernstian}\;\mathrm{slopes}=\;\mathrm{Temperature}\;\mathrm{ratio}=\frac{\mathrm{Higher}\;\mathrm{temperature}\;\left(K\right)}{\mathrm{Lower}\;\mathrm{temperature}\;\left(K\right)}$$

The experimental ratio of Nernstian slopes equals the ration between the slopes measured at high temperature to the slope measured at low temperature.

Table 2 below shows a comparison between the theoretical and experimental ratio of Nernstian slopes during the calibration of a K-ISE, using SSC and commercial reference electrodes at different temperatures (5, 14, 18, and 22 °C).

Similar results were obtained using other ISEs (Na^+^, Ca^2+^, Cl^−^, and pH) and other identical SSC reference electrodes of both IM SSC and CP SSC types. The theoretical and experimental ratio of Nernstian slopes were in good agreement for all tested reference electrodes.

### 3.6. Sample Measurements

River water samples were analyzed by direct potentiometry for all four target ions (K^+^, Na^+^, Cl^−^, and Ca^2+^) and pH using commercial and SSC reference electrodes. These measurements were carried out using CP SSC and IM SSC that were conditioned for 12 and 25 weeks, respectively. For comparison purposes, the same samples were analyzed at the Lounais-Suomen Vesi ja Ympäristötutkimus Oy (LSVY) laboratory which specialized in routine water analyses. Chloride ion (Cl^−^) was determined by ion chromatography according to the method SFS-EN ISO 10304-1. K^+^, Na^+^, and Ca^2+^ ions were determined using inductively coupled plasma optical emission spectrometry (ICP-OES) and the method SFS-EN ISO 11885. Finally, pH was determined by potentiometry according to the Method SFS 3021. Concentrations are reported as the average of three repetitions made during three consecutive days ± standard deviation except for the external laboratory results.

The results obtained for the RW1 (Aura river) and RW2 (Savijoki stream) water samples are shown in Table 3 and Table 4, respectively.

The results from similar analyses performed at 5 °C are shown in Table 5 and Table 6 below.

From the obtained results, it can be concluded that the determination of the target ions at different temperatures is possible if the calibrations are carried out at the corresponding temperatures.

## 4. Conclusions

The performance of solid-state composite (SSC) reference electrodes, manufactured by chemical polymerization (CP SSC) and injection molding (IM SSC) methods, was evaluated in this study. The stability of all the studied SSC reference electrodes was evaluated by using potentiometric and electrochemical impedance spectroscopic methods. The CP SSC and IM SSC reference electrodes obtained their full stability after 5 and 10 weeks of conditioning, respectively. Based on these results, the conditioning process of CP SSC electrodes is much faster than that for the IM SSC ones. The reason can be attributed to the difference in the used polymer and hardness and porosity of the resulting polymer/inorganic salt composite. Two so-called multi-solution protocols (MSP1 and MSP2) were used to study the influence of ionic strength, ion charge and mobility, and the pH value of the sample on the response of the SSC reference electrodes. The general conclusion is that the change in pH is the most important factor for both CP SSC and IM SSC, with a bigger effect on the IM SSC reference electrodes.

A crucial test for the suitability of SSC electrodes as reference electrode was done by using them as reference electrodes in the calibration of commercial indicator electrodes for different ions. Both CP SSC and IM SSC functioned very well as reference electrodes in these measurements.

The calibrations of commercial indicator electrodes for different ions were carried out at different temperatures using the SSC reference electrodes. The theoretical and experimental ratio of Nernstian slopes at different temperatures were in good agreement for all the tested SSC reference electrodes and comparable to the commercial reference electrode.

Determining the concentrations of different ions (K^+^, Na^+^, Cl^−^, and Ca^2+^) and pH in two river water samples was done using the SSC reference electrodes. The obtained results were in good agreement with the results obtained from measurements done by an external laboratory specialized in routine water analyses. Additionally, it was shown that the determination of the target ions at different temperatures is possible if the calibrations of the electrodes are carried out at the corresponding temperatures.

## Figures and Tables

**Figure 1 membranes-12-00569-f001:**
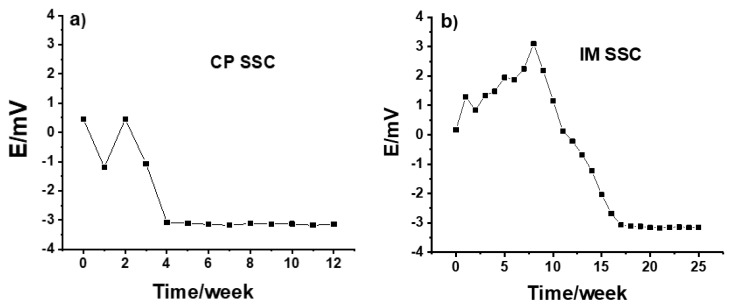
Potential stability of (**a**) CP SSC and (**b**) IM SSC electrodes measured in 10^−1^ MKCl.

**Figure 2 membranes-12-00569-f002:**
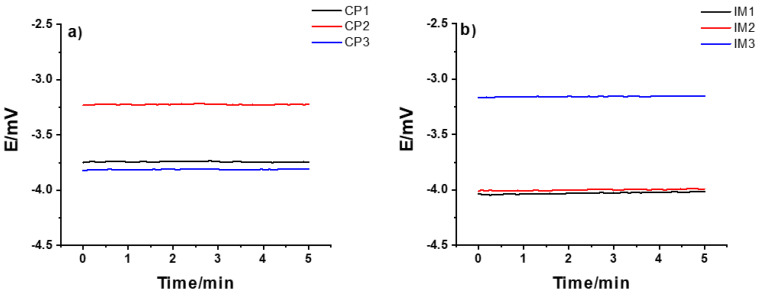
Reproducibility of the potential of (**a**) CP SSC and (**b**) IM SSC identical electrodes measured in 10^-1^ M KCl solution.

**Figure 3 membranes-12-00569-f003:**
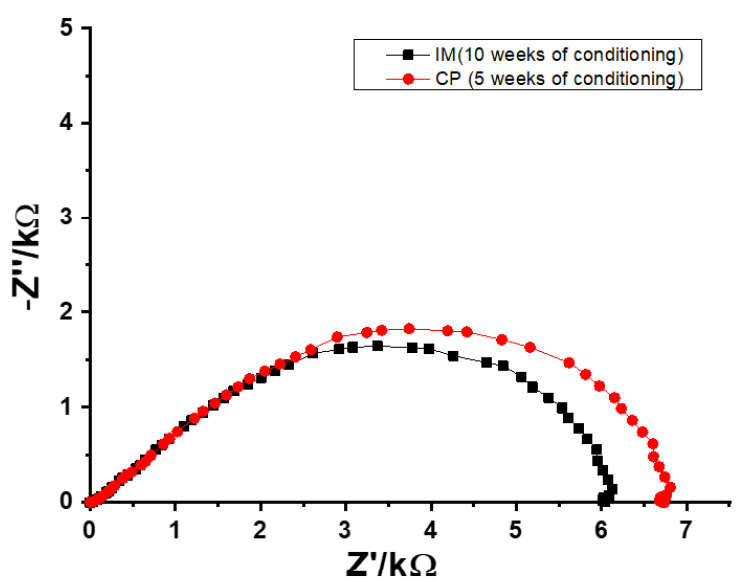
Typical impedance spectra of CP SSC and IM SSC reference electrodes recorded after 5 and 10 weeks of conditioning, respectively.

**Figure 4 membranes-12-00569-f004:**
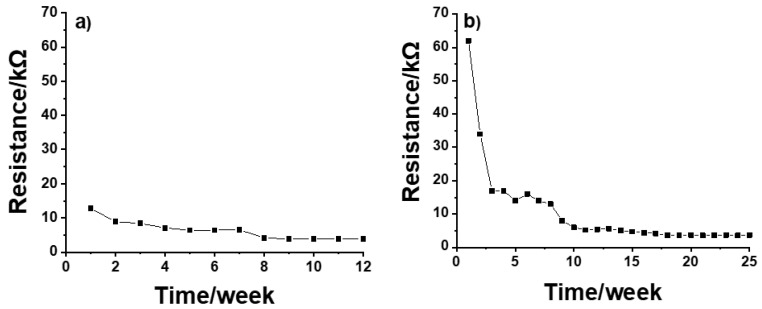
Resistance values of (**a**) CP SSC and (**b**) IM SSC reference electrodes measured during the conditioning process.

**Figure 5 membranes-12-00569-f005:**
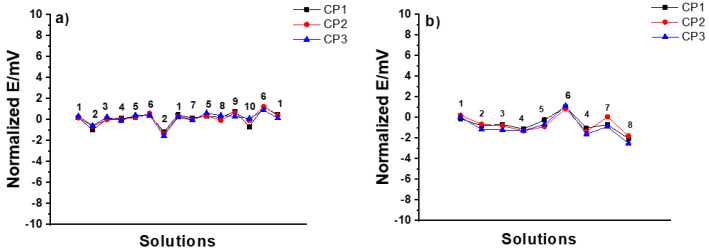
Results of (**a**) MSP1 (1: 3 M KCl, 2: DW, 3: 10^−2^ M NaCl, 4: 10^−2^ M KCl, 5: 10^−1^ M KCl, 6: 10^−2^ M HCl, 7: 10^−1^ M NaCl, 8: 10^−1^ M NaBr, 9: 10^−1^ M NaHCO_3_, 10: 10^−3^ M KOH), and (**b**) MSP2 (1: 10^−1^M KCl, 2: 10^−1^ M CaCl_2_, 3: 10^−4^ M CaCl_2_, 4: DW, 5: 10^−1^ M HCl, 6: 10^−4^ M HCl, 7: 10^−1^ M NaOH, 8: 10^−4^ M NaOH) for three identical CP SSC reference electrodes.

**Figure 6 membranes-12-00569-f006:**
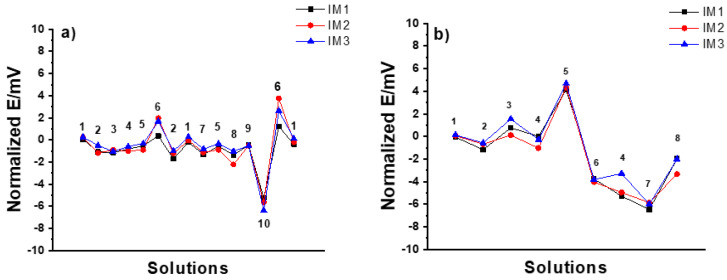
Results of (**a**) MSP1 (1: 3 M KCl, 2: DW, 3: 10^−2^ M NaCl, 4: 10^−2^ M KCl, 5: 10^−1^ M KCl, 6: 10^−2^ M HCl, 7: 10^−1^ M NaCl, 8: 10^−1^ M NaBr, 9: 10^−1^ M NaHCO_3_, 10: 10^−3^ M KOH), and (**b**) MSP2 (1: 10^−1^M KCl, 2: 10^−1^ M CaCl_2_, 3: 10^−4^ M CaCl_2_, 4: DW, 5: 10^−1^ M HCl, 6: 10^−4^ M HCl, 7: 10^−1^ M NaOH, 8: 10^−4^ M NaOH) for three identical IM SSC reference electrodes.

**Table 1 membranes-12-00569-t001:** Calibration curve parameters obtained for commercial ISEs using SSC reference electrodes. The reported values are the mean from three identical electrodes ± standard deviation (SD), and the goodness of fit represented by coefficient of determination (R^2^).

ISE	*S* ± SD (mV/Decade)		*E*^0^ ± SD (mV)		Coefficient of Determination (R^2^)	
	CP SSC	IM SSC	CP SSC	IM SSC	CP SSC	IM SSC
Potassium	58.0 ± 0.3	58.0 ± 0.2	107.6 ± 3.8	108.7 ± 1.6	(0.9986–0.9993)	(0.9989–0.9994)
Sodium	58.1 ± 0.3	58.1 ± 0.2	147.8 ± 0.5	149.6 ± 0.4	(0.9991–0.9995)	(0.9988–0.9996)
Calcium	28.2 ± 0.1	28.1 ± 0.1	101.0 ± 5.3	103.8 ± 0.1	(0.9990–0.9993)	(0.9989–0.9995)
Chloride	−56.5 ± 0.2	−56.4 ± 0.3	21.8 ± 1.4	27.6 ± 1.1	(0.9988–0.9993)	(0.9991–0.9994)
pH	57.3 ± 0.1	56.8 ± 0.1	640.4 ± 1.5	639.3 ± 0.1	(0.9998–0.9999)	(0.9997–0.9999)

The linear range for Na^+^, Ca^2+^, and Cl^−^ ISEs was 10^−1^–10^−4^ M, and for K^+^ ISE was 10^−1^–10^−4.5^ M.

**Table 2 membranes-12-00569-t002:** Theoretical and experimental ratio of Nernstian slopes at different temperatures in the calibration of K-ISE. The values of the slopes in mV/decade are given in parenthesis.

Temperature Change	From 5 °C to 22 °C	From 14 °C to 22 °C	From 18 °C to 22 °C
Theoretical Nernstian ratio	1.06 (55.19, 58.57)	1.03 (56.98, 58.57)	1.01 (57.77, 58.57)
Experimental Nernstian ratio IM1	1.05 (55.6, 58.2)	1.03 (56.3, 58.2)	1.01 (57.5, 58.2)
Experimental Nernstian ratio CP3	1.05 (55.4, 58.3)	1.04 (56.1, 58.3)	1.02 (56.9, 58.3)
Experimental Nernstian ratio Orion	1.04 (55.4, 57.7)	1.03 (56.0, 57.7)	1.02 (56.5, 57.7)

**Table 3 membranes-12-00569-t003:** Ion concentrations measured in RW1 at room temperature (22 ± 1 °C) using SSC and commercial reference electrodes and compared with the results from routine laboratory analyses.

Reference Electrode	K^+^ Concentration (mg/L)	Na^+^ Concentration (mg/L)	Cl^−^ Concentration (mg/L)	Ca^2+^ Concentration (mg/L)	pH
IM1	1.61 ± 0.05	7.06 ± 0.24	9.62 ± 0.13	9.73 ± 0.25	7.46 ± 0.00
IM2	1.59 ± 0.02	7.10 ± 0.16	9.56 ± 0.09	9.54 ± 0.15	7.47 ± 0.01
IM3	1.59 ± 0.04	7.13 ± 0.31	9.45 ± 0.22	9.39 ± 0.13	7.46 ± 0.01
CP1	1.60 ± 0.04	7.13 ± 0.18	9.65 ± 0.13	9.50 ± 0.26	7.53 ± 0.01
CP2	1.59 ± 0.03	7.21 ± 0.11	9.57± 0.18	9.73 ± 0.07	7.48 ± 0.01
CP3	1.62 ± 0.03	7.10 ± 0.13	9.52 ± 0.10	9.67 ± 0.25	7.48 ± 0.00
Commercial	1.60 ± 0.02	7.14 ± 0.27	9.62 ± 0.14	9.51 ± 0.19	7.51 ± 0.01
External laboratory	1.60	7.20	9.10	9.30	7.50

**Table 4 membranes-12-00569-t004:** Ion concentrations measured in RW2 at room temperature (22 ± 1 °C) using SSC and commercial reference electrodes and compared with the results from routine laboratory analyses.

Reference Electrode	K^+^ Concentration (mg/L)	Na^+^ Concentration (mg/L)	Cl^−^ Concentration (mg/L)	Ca^2+^ Concentration (mg/L)	pH
IM1	2.06 ± 0.11	10.05 ± 0.21	9.84 ± 0.16	13.72 ± 0.33	7.72 ± 0.02
IM2	1.99 ± 0.05	10.27 ± 0.26	9.74 ± 0.13	13.73 ± 0.14	7.71 ± 0.02
IM3	2.02 ± 0.05	10.24 ± 0.13	9.73 ± 0.38	13.40 ± 0.13	7.72 ± 0.02
CP1	2.07 ± 0.04	10.16 ± 0.34	9.76 ± 0.12	13.53 ± 0.41	7.73 ± 0.01
CP2	2.07 ± 0.01	10.32 ± 0.27	9.78 ± 0.10	13.15 ± 0.48	7.69 ± 0.02
CP3	2.08 ± 0.04	10.15 ± 0.32	9.84 ± 0.22	13.67 ± 0.23	7.72 ± 0.02
Commercial	2.13 ± 0.01	9.99 ± 0.49	9.91 ± 0.23	13.70 ± 0.14	7.69 ± 0.00
External laboratory	2.10	10.00	10.00	14.00	7.70

**Table 5 membranes-12-00569-t005:** Ion concentrations measured in RW1 at 5 °C using SSC and commercial reference electrodes and compared with the results from routine laboratory analyses measured at room temperature.

Reference Electrode	K^+^ Concentration (mg/L)	Na^+^ Concentration (mg/L)	Cl^−^ Concentration (mg/L)	Ca^2+^ Concentration (mg/L)	pH
IM1	1.61 ± 0.03	7.18 ± 0.16	9.58 ± 0.31	9.53 ± 0.19	7.77 ± 0.03
IM2	1.62 ± 0.05	7.17 ± 0.16	9.57 ± 0.09	9.75 ± 0.05	7.80 ± 0.03
IM3	1.61 ± 0.01	7.32 ± 0.18	9.42 ± 0.11	9.78 ± 0.13	7.81 ± 0.01
CP1	1.62 ± 0.01	7.58 ± 0.12	9.46 ± 0.17	9.78 ± 0.07	7.80 ± 0.02
CP2	1.61 ± 0.01	7.36 ± 0.35	9.65 ± 0.31	9.63 ± 0.25	7.80 ± 0.02
CP3	1.60 ± 0.02	7.34 ± 0.28	9.57 ± 0.19	9.50 ± 0.13	7.77 ± 0.01
Commercial	1.60 ± 0.02	6.80 ± 0.44	9.60 ± 0.08	9.42 ± 0.19	7.78 ± 0.01
External laboratory	1.60	7.20	9.10	9.30	7.50

**Table 6 membranes-12-00569-t006:** Ion concentrations measured in RW2 at 5 °C using SSC and commercial reference electrodes and compared with the results from routine laboratory analyses measured at room temperature.

Reference Electrode	K^+^ Concentration (mg/L)	Na^+^ Concentration (mg/L)	Cl^−^ Concentration (mg/L)	Ca^2+^ Concentration (mg/L)	pH
IM1	2.01 ± 0.03	9.25 ± 0.14	9.85 ± 0.18	13.60 ± 0.26	8.11 ± 0.03
IM2	1.97 ± 0.01	9.27 ± 0.30	9.99 ± 0.19	13.64 ± 0.13	8.10 ± 0.02
IM3	2.01 ± 0.03	9.33 ± 0.32	9.91 ± 0.26	13.42 ± 0.10	8.09 ± 0.02
CP1	2.00 ± 0.03	9.75 ± 0.11	9.81 ± 0.21	13.32 ± 0.38	8.13 ± 0.01
CP2	2.03 ± 0.02	9.47 ± 0.15	9.72 ± 0.14	13.57 ± 0.42	8.09 ± 0.01
CP3	2.01 ± 0.01	9.59 ± 0.31	9.84 ± 0.17	13.73 ± 0.22	8.08 ± 0.02
Commercial	1.97 ± 0.02	9.55 ± 0.26	9.84 ± 0.26	13.16 ± 0.16	8.10 ± 0.01
External laboratory	2.10	10.00	10.00	14.00	7.70

## Data Availability

Slim Blidi Master Thesis: Doria.

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
