# Peer review of "Long-Time Evaluation of Solid-State Composite Reference Electrodes"

_membranes, 2022, doi:10.3390/membranes12060569_

Round 1

Reviewer 1 Report

Comments:

This work presents a detail characterization of solid-state composite reference electrodes with detail results and discussion. The idea is fair and results are very important in this field. Some comments are given to authors to further improve the quality before publication. Current decision is “minor revision.

  1. In the introduction, authors are suggested to provide more information about the comparison of various methods of reference electrode for common readers. Especially the detail of chemical polymerization should be given including the mechanism and process concept.
  2. The quality of figures should be improved for smoothly read. For example, there are two (a) and (b) marked in Fig. 1, 2, 4, 5, 6. The wording in figures can be enlarged. The group name is also suggested to put in the figure for quick and distinguish information between 2 similar figures.
  3. As shown in Section 3.4, the detail information of ion selective electrodes should be given.
  4. The raw data for Table 2 are suggested to provide since common reader may need this important information.
  5. For the temperature effect, authors are suggested to provide temperature higher than 22oC (e.g., 30, 40, 50, 60, 70, 80oC).
  6. Measurement results of fabricated reference electrodes shown in section 3.6 are suggested to provide more information of various ion concentration detection including the type of ion selective electrodes (e.g., commercial product with company and country). It is confused that reference electrode can not be used to measure different ion concentration solely.

Reviewer 2 Report

The manuscript describes thorough evaluation of long-time performance of two types of solid state Ag/AgCl reference electrodes: one manufactured by chemical polymerization and the other by injection moulding. Stability was monitored over several months and additionally the effect of sample composition, concentration and pH was evaluated. Lastly, performance of the electrodes was evaluated with several commercial ISEs, both in calibration experiments and real water samples. In all experiments the tested electrodes showed very good results, with a somewhat larger effect on sample pH on the injection moulded electrodes. 

Such long term stability of novel reference electrodes is seldom evaluated, yet very important, which makes this work valuable for sensor chemists planning continuous measurements over long periods of time. The manuscript is very well written, results presented clearly, and I would like to see this paper published after the following minor corrections are taken into account: 

- The title states "long-time evaluation", yet it is not clear from the manuscript at what point in time were the calibrations and river water measurements conducted. Was this done after 12/25 weeks of conditioning? Please explain this in the manuscript. Alternatively, if these were done right after electrode manufacture, I suggest adding this to the title. E.g. : "Performance and long-time evaluation of..." 

-How were the linear ranges in table 1 determined? Please explain this in section 2.3.4. and preferably give R2 values in the results.

- Please refer to Figure 4 somewhere in the text, I suppose line 195.

Reviewer 3 Report

The manuscript under review is devoted to the characterization of the performance of two types of solid state composite reference electrodes. The topic is timely and definitely worth of studying. The described electrode evaluation protocol appears to be relevant. The paper is written in a clear and consistent way, the results are convincing and I have really enjoyed the reading. Overall, I believe this report can be accepted for publication in Membranes.

However, I have a number of questions to the authors and their clarification is very important to improve the clarity and consistency of the paper:

1) Neither the original ref. [19] nor the manuscript under review describes the ratio between the components of the mixture for photo polymerization (PVAc, VAc, KCl, DMPP) – this leaves very subtle chances for someone to reproduce these studies. The authors have  to provide the details on the employed composition. The same holds for the composition of reference electrode produced by injection moulding (ref. [21]).

2) Ref 19 describes 2 compositions (SSC1 and SSC2) – which particular was employed here? No LiOAc was added as far as I can understand.

3) The conditioning time of the prepared SSC before starting the experiments are not given in the experimental section.

4) It is not clear from the experimental part why the period of study was different for CP and IM electrodes (12 and 25 weeks).

5) The motivation behind the particular choice of cations and anions for MSP is not provided. Would it be possible to include more lipophilic anions in this protocol, like e.g. SCN-? Could the authors comment on the influence of ion lipophilicity on the performance of SSC-RE?

6) What were the reasons to introduce two different MSPs?

In my opinion the manuscript has to be revised to address these concerns.
